# Testing models of reciprocal relations between social influence and integration in STEM across the college years

**Paul R. Hernandez** [1]*, **V. Bede Agocha**[2], **Lauren M. Carney**[2], **Mica Estrada**[3], **Sharon Y. Lee**[2], **David Loomis**[4], **Michelle Williams**[2], **Crystal L. Park**[2]

**1** Department of Teaching, Learning, and Culture, Texas A&M University, College Station, Texas, United States of America, **2** Department of Psychological Sciences, University of Connecticut, Storrs, Connecticut, United States of America, **3** Department of Social and Behavioral Sciences, University of California San Francisco, San Francisco, California, United States of America, **4** Department of Learning Sciences, West Virginia University, Morgantown, West Virginia, United States of America

* prhernandez@tamu.edu

**Data Availability Statement:** All data and meta-data files are available from the online Dryad database (doi:10.5061/dryad.k98sf7m3k.)

## Abstract

The present study tests predictions from the Tripartite Integration Model of Social Influences (TIMSI) concerning processes linking social interactions to social integration into science, technology, engineering, and mathematics (STEM) communities and careers. Students from historically overrepresented groups in STEM were followed from their senior year of high school through their senior year in college. Based on TIMSI, we hypothesized that interactions with social influence agents (operationalized as mentor network diversity, faculty mentor support, and research experiences) would promote both short- and long-term integration into STEM via social influence processes (operationalized as science self-efficacy, identity, and internalized community values). Moreover, we examined the previously untested hypothesis of reciprocal influences from early levels of social integration in STEM to future engagement with social influence agents. Results of a series of longitudinal structural equation model-based mediation analyses indicate that, in the short term, higher levels of faculty mentorship support and research engagement, and to a lesser degree more diverse mentor networks in college promote deeper integration into the STEM community through the development of science identity and science community values. Moreover, results indicate that, in the long term, earlier high levels of integration in STEM indirectly influences research engagement through the development of higher science identity. These results extend our understanding of the TIMSI framework and advance our understanding of the reciprocal nature of social influences that draw students into STEM careers.

## Introduction

National attention has focused on the need to attract, retain, and adequately prepare a larger and more diverse science, technology, engineering, and mathematics (STEM) workforce to accelerate innovation and discovery, maintain global competitiveness, and expand economic

**Funding:** C. Park and M. Williams. This project was funded was provided by National Institute of General Medical Sciences (Grant No. 1R01GM107707). https://www.nigms.nih.gov/ The funder had no role in study design, data collection and analysis, decision to publish, or preparation of the manuscript.

**Competing interests:** The authors have declared that no competing interests exist.

prosperity [1–8]. Research on and projections of national trends indicate STEM-related occupations will grow at nearly twice the rate of non-STEM occupations, and most of these occupations will require post-secondary education and training [4, 7, 9, 10]. However, reports and research on national post-secondary STEM education attainment also point to a continuing shortfall due to such factors as student flight from STEM majors and college dropout–particularly in the first two years of college [3, 11, 12].

## The rise of mentorship as an antidote to attrition

There is growing interest and research on the roles that mentors (and other socializing experiences, such as undergraduate research) play in supporting learning and degree attainment in college [13–15]. There is ample cross-sectional and qualitative research on mentorship in college contexts; however, robust longitudinal or experimental data showing the impact of mentorship on student success and persistence is scant [16–21]. For example, a number of cross-sectional correlational studies with undergraduate students in a variety of disciplines have examined quantitative or qualitative associations between mentorship support and (a) psychosocial variables (e.g., academic and social integration, depression and stress, motivation), (b) academic success, and (c) persistence in college [22–37]. Overall, systematic reviews of the mentoring literature in college contexts indicate only modest associations between having a mentor in college and learning or persistence, and small-to-moderate associations between the specific types of mentorship support received (e.g., psychosocial-emotional support) and psychosocial outcomes (e.g., motivation) [38, 39].

Social psychology theoretical models–with their emphasis on person-by-situation interactions–are useful for describing and explaining the kinds of interactions that promote internal motivational processes that, in turn, promote learning or persistence in STEM [13, 20, 40]. Social psychology theories, such as Social Cognitive Career Theory (SCCT) [41], are increasingly used to better understand why, for whom, and under what circumstances mentorship influences outcomes in higher education [15–20]. For example, SCCT, an extension of Social Cognitive Theory [42, 43], posits that prior performance attainment (e.g., prior success in a domain such as science), vicarious learning (e.g., inspiration or learning drawn from observing role models), social persuasion (e.g., realistic encouragement from a mentor), and emotional arousal (e.g., tension) are key determinants of self-efficacy (i.e., confidence in one's ability to successfully execute specific tasks) and outcome expectations (i.e., one's appraisal of the likely outcome of executing those tasks). And efficacy and outcome expectations, in turn, play a central in the development and pursuit of career interests and future performance attainments, and performance attainment then becomes a source for future efficacy and outcome expectation development [41]. A few scholars have used SCCT (and other social psychology theories) to test hypotheses concerning mentoring and persistence in STEM disciplines, particularly for students from historically underrepresented (HU) minority groups in STEM (e.g., racial minorities, women). In general, theoretically driven research in STEM contexts has found more consistent and somewhat larger associations between mentoring and STEM-related psychosocial or persistence outcomes than has been reported in studies of mentoring in more heterogeneous college contexts. For example, researchers using SCCT [41, 44] have found moderate positive associations between mentorship support and psychosocial outcomes (e.g., science self-efficacy), as well as small-to-moderate associations with intentions to persistence in STEM among HU college students (i.e., racial/ethnic minorities) [45–48]. Similarly, researchers studying women pursuing STEM degrees have shown significant small-to-moderate positive associations between mentorship and a sense of belong, science identity, and intentions to persist in a scientific career [49, 50]. However, even theoretically driven research

has not typically examined longitudinal associations linking mentoring, motivational media-tors, and integration into STEM, nor has this research tested reciprocal relations or feedback loops.

## Social influence model of attracting and retaining aspiring STEM professionals

The approach taken in the current study is to frame mentoring in STEM within a social influ-ence model whereby the mentor takes the role of influencer and the undergraduate student is the subject of influence. This approach grows out of a body of research showing that social influence occurs continuously and across contexts [51], including educational settings where higher education students integrate or disengage from their disciplinary communities [52]. Theory and empirical research indicate that scientific community members (e.g., mentors) and relevant scientific training environments (e.g., research experiences) could operate as socializing or social influence agents [31, 53–56]. Scientific mentors, in particular, may be powerful influencers because of their ability to confer or withhold valued rewards (such as grades, degrees, etc.), which are desirable to members of the community. Mentors also hold high relational value because they can provide access to both desirable social and material out-comes such as affiliation, support, resources, and more opportunities in life [57]. These attri-butes undoubtedly make mentors powerful influencers of their mentees, as they support students' integration into their disciplinary community. What, however, does it mean to inte-grate into a social group or community?

**Social influence and student integration.**   Herbert Kelman proposed and experimentally tested a social influence model, which predicted the conditions of socializing an individual into a new role or resocialization that moves individuals from old to new roles [58–61]. Kel-man's Tripartite Integration Model of Social Influence (TIMSI) framework posits that persons and social contexts can operate as social influence agents. Influence agents integrate the targets of social influence into a social system by rewarding compliance with social norms (rule orien-tation), cuing the role by which the target-to-influence is identified (role orientation), and/or reinforcing how the groups' values are consistent with the target of influence's internalized val-ues (values orientation). The result of successful influence is greater efficacy to engage in nor-mative behaviors, stronger identification with the influencing agent's group, and internalization of that group's values. And the social influence processes (i.e., efficacy, identity, and internalized values), in turn, promote deeper integration into the community. Further-more, Kelman readily acknowledged that the direction of influence could be reciprocal–partic-ularly in long-term relationships, as opposed to experimentally manipulated brief social interactions between strangers [61]. That is, reciprocal social influence describes the process whereby individuals who desire to integrate more deeply into a social system engage in activi-ties that further integrates them into the social system. An example of this reciprocal process in the STEM academic context would be when faculty mentorship and research experiences (that increase student science identity) leads to students' persistence, and then those students with higher intentions reciprocally are more likely to engage in additional research experi-ences. In this way, students are not passive recipients of influence, but also active agents in seeking out further opportunities to be influenced.

**Agent influence is mediated by social influence.**   Kelman's theory, the TIMSI framework, has been useful to better understand how students integrate into their disciplinary communi-ties [62]. A small but growing body of evidence demonstrates how TIMSI predictions concern-ing the relationships between social influence agents, social influence processes, and social integration among college students in STEM disciplines occur (see Fig 1; Social Influence

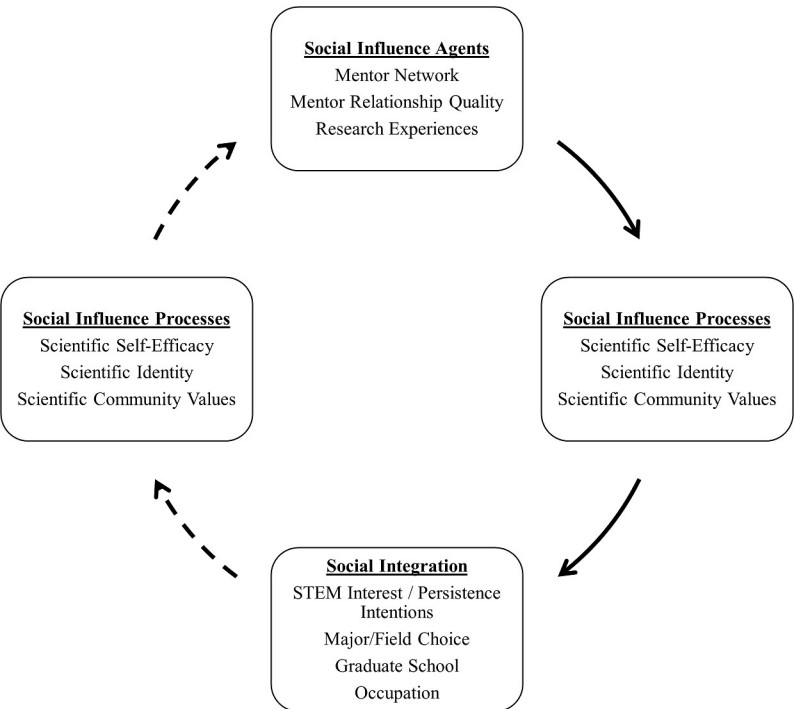

**Fig 1. Conceptual model of the Tripartite Integration Model of Social Influences (TIMSI).** Solid-line arrows represent theorized influence from influence agents on socialization or resocialization through influence orientations. Dashed-line arrows represent theoretically plausible reciprocal feedback loops.

Agents → Social Influence Processes → Social Integration). Several recent studies have investigated the effects of social influence agents, such as mentors, on social influence processes and down-stream social integration in STEM contexts. For example, the most comprehensive test of TIMSI predictions tracked the effects of faculty mentorship quality and the number of semesters engaged in research experiences in college on changes in science self-efficacy, identity, and values, as well as distally measured engagement in or dropout from STEM disciplines over a six-year period (junior year of college through four years post-baccalaureate attainment) in a national sample of HU undergraduate STEM majors [63]. These researchers operationalized Kelman's model by measuring the social influence processes in terms of science self-efficacy, science identity, the endorsement of science community values, and post-baccalaureate persistence in a scientific career (social integration). The study revealed that (a) higher levels of faculty mentorship quality and higher engagement in research experiences were associated with higher levels of scientific self-efficacy, scientific identity, and scientific community values, (b) higher levels of scientific identity were associated with a higher probability of persistence in a STEM career after completing an undergraduate degree, and (c) mentoring and research experiences had a positive indirect effect on post-baccalaureate STEM career choices through the development of science identity as an undergraduate.

Additional studies have examined TIMSI predictions in a more limited fashion, testing TIMSI predictions concerning the relationships between social influence agents, social influence processes, and social integration among college students over a shorter period of time (e.g., a 10-week summer research experience), or focusing more narrowly on TIMSI predictions concerning the relationships between social influence processes and social integration (Social Influence Processes → Social Integration) [62, 64, 65]. The results from these more

limited studies provide convergent evidence indicating higher levels of mentorship support lead to higher levels of the efficacy, identity, and values (i.e., social influence processes) and higher levels of social influence processes, particularly science identity, lead to higher levels of persistence [62, 64, 65].

**Limitations of the extant research.** Although emerging evidence for the TIMSI framework is promising, key aspects of the model have yet to be rigorously tested. First, studies have not examined the degree to which TIMSI operates similarly (or differently) for identifiable demographic subgroups in STEM. To date, the most robust longitudinal evidence for TIMSI was derived from a national sample of HU racial/ethnic minority students in STEM. Less is known about the longitudinal TIMSI process for students from HO groups in STEM, or for men versus women in STEM. Second, the model has not been tested across a variety of critical time points in the academic career. For example, longitudinal mediation studies have primarily focused on students in the final two years of college and have only examined the indirect effects of social influence agents on integration through social influence processes [63, 65]. National trends indicate that most departures from STEM majors occur during the first two years of college. Thus, testing TIMSI predictions at earlier points in college may reveal theoretically meaningful shifts in the processes by which students integrate into STEM majors and career tracks.

Third, prior tests of the TIMSI prediction linking mentoring to social influence processes and integration operationalized mentoring as the level of support received from a single (typically faculty) mentor. However, advances in mentorship theory have postulated that developing a diversified network of mentors (i.e., diversity in roles [faculty, post baccalaureate, more advanced undergraduate, peer]) can provide mentees with more comprehensive support for their learning and career development [15, 66–72]. Mentor network theory suggests individuals benefit from having a diversified network because they are able to rely on the specific strengths of individual mentors serving in a specific supportive role rather than relying on a single mentor providing support across multiple roles. Although there is scant longitudinal evidence for the theorized link between mentor network characteristics and social influence processes, cross-sectional research indicates that undergraduate STEM majors with larger and more diverse networks of scientific mentors report higher levels of scientific self-efficacy and identity [66, 67, 73]. Thus, testing TIMSI predictions where mentorship is operationalized in terms of both the quality of support from a faculty mentor and mentor network diversity may reveal theoretically important information about the kinds of support that lead to social influence.

Fourth, although the TIMSI framework posits the potential for reciprocal influence over longer periods of time (as opposed to short-term manipulated experiences), no research has yet examined the theoretically plausible reciprocal mediated effects of social integration on future engagement with social influence agents through the social influence process (see dashed lines in Fig 1; Social Integration → Social Influence Processes → Social Influence Agents). For example, students who feel more socially integrated may more actively seek out mentors, grow their mentor networks, and pursue professional development experiences that in turn deepen their social integration. In this way, there could be a self-perpetuating cycle that deepens students' integration into the scientific community.

## Current study

The current study tests predictions from the TIMSI framework in the context of college students pursuing a STEM degree and addresses several limitations in the extant literature. In a longitudinal study of college students in STEM disciplines, we hypothesized that the level of

faculty mentor support, diversified college-related mentor networks, and research experiences in college would positively influence the development of science self-efficacy, science identity, and the internalization of science community values. Further, we expected that science self-efficacy, identity, and values would, in turn, promote scientific career persistence intentions (i.e., integration into the scientific community).

The present study also strategically extends the extant literature and theory. First, much of the longitudinal research on the TIMSI framework tested predictions among HU students in STEM [62, 63]. The present study focuses on the experiences of HO undergraduates in STEM majors to better understand and test TIMSI hypotheses. That is, prior longitudinal research excluded HO students and the present study is an opportunity to describe the socialization experiences of these students. In addition, the present study draws on data from a larger study that included a relatively small number of HU students in STEM. Given the relatively small numbers HU STEM students and our focus on longitudinal analyses in a structural equation modeling framework (more details provided below), we were unable to estimate models of HU-HO group differences (i.e., tests of measurement invariance and multiple groups SEM were not possible). Thus, the present study has the potential to identify subgroup similarities or differences in the patterns of results based on majority status in comparison to prior studies (but not as a direct comparison within the present study).

Second, longitudinal research on the TIMSI framework has primarily focused on students in the last two years of college (i.e., junior and senior years) or in graduate school [62, 63, 65]. The current study, by contrast, follows students from the spring semester prior to college (i.e., rising first year college students) through the spring semester of their fourth year of college (i.e., senior year). Thus, the present study has the potential to identify developmental changes or equilibrium as students matriculate from high school throughout their undergraduate tenure. Third, prior longitudinal research primarily focused on the influence faculty mentors have on their undergraduate mentee's integration into STEM. The present study was designed to address the unique influence of faculty mentorship quality and mentor network diversity on student integration into STEM.

Fourth, to date longitudinal tests of the TIMSI framework have only focused on the process by which social influence agents impact future integration into STEM [62, 63, 65], ignoring the theoretically plausible reciprocal influence of social integration on future engagement with influence agents. Notably, it may be unreasonable to ignore reciprocal effects when investigating causal ordering or longitudinal mediation processes that involve humans interacting in a social environment [74–76]. Given that research findings in social psychology documenting how reciprocity contributes significantly to the forming and maintaining of a variety of social relationships [77], it is reasonable to hypothesize that mentees and mentors exist in relationship to each other, with reciprocal influences occurring [20]. To test this notion, the present longitudinal mediation study simultaneously examined the process by which social influence agents affect present and future social integration into STEM, as well as the reciprocal process by which social integration in STEM affects future engagement with social influence agents. However, investigations of longitudinal mediation (or causal ordering) with contemporaneous and reciprocal effects involve relatively complex models with many potential paths of direct and indirect influence. A primary goal in these types of longitudinal studies is to first identify the simplest model that provides adequate fit to the data before examining mediated effects. Best practices in longitudinal causal ordering research suggest that investigations compare an *a priori* complex model that allows all possible forward pathways to a set of *a priori* successively simpler or more constrained models [74–76, 78, 79]. Consistent with best practices, we adopted this model comparison approach and conducted a series of eight nested SEMs to systematically test for longitudinal, contemporaneous, and reciprocal relations among the

outcomes, mediators, and predictors in our model (see supplemental materials for a complete discussion of Models 1–8; S1 and S2 Figs).

In summary, the present study addresses three research questions to test predictions from and extend knowledge about the TIMSI framework in the context of STEM education. First, do social influence processes mediate the effect of social influence agents on social integration into STEM over an undergraduates' tenure in college? Second, do social influence processes mediate the effect of social integration in STEM on future engagement with social influence agents over the undergraduates' tenure in college? Third, do the mediated effects operate longitudinally, contemporaneously, or both?

## Materials and methods

### Participants

All study procedures were approved by the University of Connecticut institutional review board (#H14-213WVU). Written informed consent was obtained from all study participants. The current study was drawn from a larger longitudinal study ($N$ = 1,839) of student development in the graduating class of 2019 at a research-intensive (i.e., Carnegie classification of "very high research activity"), four-year public land-grant university located in the northeastern U.S. The larger study focused on the development cognitive, emotional, and behavioral self-regulation among all students (regardless of major) over the college tenure. However, students in non-STEM majors were not asked questions about their scientific persistence intentions (the primary outcome of this study) in an effort to reduce fatigue and mitigate potential reactivity to irrelevant survey questions. Historical enrollment data at the university indicate that university undergraduates are 51% female and 49% male, 56% White, 11% Asian, 11% Hispanic, 10% foreign nationals, 6% Black, 3% multi-racial, and 3% undeclared race/ethnicity [80]. Prior research in the larger longitudinal study indicated that study participants largely mirrored the demographics of the university, with the exception of somewhat smaller proportion of students who chose not to declare their race, African American students, and Hispanic students [81]. However, the present study focuses on a subsample of the larger sample, that is, HO students in STEM majors.

The analytic sample for the present study consists of 751 first-year HO students that declared a STEM major over their college tenure. STEM majors included those related to science (i.e., agricultural, biological/life, physical), technology (e.g., computer science), engineering, mathematics (e.g., actuarial, mathematics, statistics), and pre-professional medicine (e.g., pre-pharmacology, nursing, nutrition science). As noted above, only students that self-identified as being in a STEM major were asked questions about their intention to persist in a scientific career. Of the larger sample, 589 were excluded from analyses, because they were non-STEM majors, 243 were non-first-year students (e.g., transfer), 162 were from HU groups or did not respond to demographic questions, 56 did not respond to the surveys after giving consent, and 38 were excluded due to academic dismissal from the university.

At the time of recruitment, in the spring and summer prior to the first-year of college, 58% of the analytic sample ($N$ = 751) self-identified as female and 42% as male (Table 1). For race and ethnicity, 76% of the majority students in the analytic sample self-identified as White and 24% as Asian (Table 1). Concerning family background, 22% of the sample that responded to the parental education questions ($n$ = 572 of 751) indicated that they were first-generation in their family to attend college (i.e., reported that neither their mother nor father had earned a baccalaureate degree or higher), and the average family annual household income was between

**Table 1. Summary of sample descriptive statistics.**

| Variables | Time | N | M | SD | Skew | Kurtosis |
|---|---|---|---|---|---|---|
| Female (0 = male, 1 = female) | T1 | 751 | 0.58 | 0.49 | -0.30 | -1.90 |
| First-generation (0 = not, 1 = First Gen.) | T1 | 572 | 0.22 | 0.42 | 1.34 | -0.20 |
| Family income | T1 | 583 | 4.66 | 1.82 | -0.18 | -0.36 |
| First semester cumulative GPA | Fall Year 1 | 723 | 3.28 | 0.63 | -1.28 | 2.01 |
| STEM Career Persistence Intentions | T1 | 613 | 7.95 | 1.78 | -1.09 | 0.71 |
| STEM Career Persistence Intentions | T2 | 463 | 7.55 | 1.95 | -0.95 | 0.81 |
| STEM Career Persistence Intentions | T2 | 548 | 7.62 | 1.97 | -0.99 | 0.65 |
| STEM Career Persistence Intentions | T3 | 548 | 8.09 | 1.97 | -0.85 | 0.27 |
| STEM Career Persistence Intentions | T4 | 463 | 7.60 | 2.02 | -0.78 | -0.08 |
| Science Self-Efficacy | T1 | 618 | 3.85 | 0.64 | -0.34 | 0.24 |
| Science Self-Efficacy | T2 | 463 | 3.69 | 0.80 | -0.26 | -0.10 |
| Science Self-Efficacy | T3 | 548 | 3.73 | 0.74 | -0.36 | 0.37 |
| Science Self-Efficacy | T4 | 549 | 3.78 | 0.73 | -0.30 | 0.13 |
| Science Self-Efficacy | T5 | 463 | 3.87 | 0.77 | -0.59 | 0.55 |
| Science Identity | T1 | 617 | 3.67 | 0.87 | -0.59 | 0.15 |
| Science Identity | T2 | 462 | 3.58 | 0.85 | -0.29 | 0.09 |
| Science Identity | T3 | 547 | 3.67 | 0.87 | -0.48 | -0.04 |
| Science Identity | T4 | 549 | 3.62 | 0.90 | -0.30 | -0.43 |
| Science Identity | T5 | 462 | 3.62 | 0.93 | -0.50 | -0.24 |
| Scientific Community Values | T1 | 613 | 5.08 | 0.94 | -1.33 | 1.71 |
| Scientific Community Values | T2 | 460 | 4.89 | 0.99 | -0.97 | 0.95 |
| Scientific Community Values | T3 | 548 | 4.98 | 0.95 | -1.09 | 1.22 |
| Scientific Community Values | T4 | 548 | 4.24 | 0.65 | -1.19 | 2.80 |
| Scientific Community Values | T5 | 462 | 4.13 | 0.75 | -1.04 | 0.93 |
| Faculty Mentorship Support | T2 | 136 | 3.36 | 0.76 | -0.03 | 0.25 |
| Faculty Mentorship Support | T3 | 224 | 3.47 | 0.69 | -0.32 | 1.05 |
| Faculty Mentorship Support | T4 | 242 | 3.55 | 0.69 | -0.25 | 0.79 |
| Faculty Mentorship Support | T5 | 256 | 3.64 | 0.75 | -0.43 | 0.30 |
| Mentor Network Diversity | T1 | 619 | 2.74 | 1.88 | 0.21 | -0.63 |
| Mentor Network Diversity | T2 | 516 | 1.48 | 1.39 | 0.98 | 0.42 |
| Mentor Network Diversity | T3 | 614 | 1.67 | 1.48 | 0.62 | -0.44 |
| Mentor Network Diversity | T4 | 613 | 1.85 | 1.59 | 0.53 | -0.65 |
| Mentor Network Diversity | T5 | 549 | 1.93 | 1.63 | 0.52 | -0.72 |
| Research Experiences | T2 | 475 | 0.72 | 1.15 | 2.52 | 8.75 |
| Research Experiences | T3 | 568 | 1.15 | 1.44 | 1.66 | 3.10 |
| Research Experiences | T4 | 569 | 1.53 | 1.74 | 1.46 | 2.31 |
| Research Experiences | T5 | 549 | 1.85 | 1.99 | 1.04 | 0.35 |

M = mean. N = sample size (cases with complete data for a given variable). SD = standard deviation. T1 = pre-college, T2 = spring 1st year of college, T3 = spring 2nd year of college, T4 = fall 3rd year of college, T5 = spring 4th year of college. Concerning race/ethnicity, $n$ = 177 self-identified as Asian and $n$ = 574 self-identified as White; Family income was an ordinal variable coded 1 = <$30k, 2 = $30k-50k, 3 = <$50k-$75k, 4 = $75k-$100k, 5 = $100k-$150k, 6 = $150k-$200k, 7 = $200k-$250k, 8 = >$250k.

$75,000-$100,000 per year (Table 1). Finally, with respect to performance, according to university administrative records the average first-semester of college cumulative grade point average was 3.24 on a 4.0 scale (Table 1).

## Procedure

The incoming cohort of students accepted for fall 2015 admissions to the university was recruited to participate in this study. Study recruitment involved (a) a study flyer included in the official university orientation materials emailed to students in the spring prior to college, (b) announcements during university orientation sessions in the summer prior to college, and (c) email invitations and reminders sent to students accepted to the university. In the spring and summer of 2015 (i.e., prior to the first-year of college), participants completed an informed consent form, as well as a two-part online survey, which included the ACT ENGAGE survey (i.e., a self-report assessment typically used by colleges to help identify students at risk of academic struggles in college) and a custom-designed research survey administered in Qualtrics.

As part of the longitudinal design, participants completed follow-up online surveys each semester, and university administrative data were gathered to track student progress toward degree attainment (e.g., academic standing, declared major, courses taken, grade point average). Participants received nominal compensation ($20) for completing each survey.

## Measures

All scales and checklists were administered via self-report in online surveys. All reliability estimates and psychometric tests (e.g., CFAs, tests of measurement invariance) are included in the supplemental materials and S1 and S2 Tables. Unless otherwise noted, scales were administered on five occasions from high school through the fourth year of college (a) T1 is spring and summer prior to college (2015), (b) T2 is spring first year of college, (c) T3 is spring second year of college, (d) T4 is fall third year of college, and (e) T5 is spring fourth year of college.

**Scientific career persistence intentions.** The primary social integration outcome, scientific persistence intention, was measured using a three-item scale [82]. Participants read the following items: "To what extent do you plan to pursue a science-related research career?," "What is the likelihood of you obtaining a science-related degree?," and "What is the likelihood of you applying to graduate school?" Participants rated the strength of their intentions on a scale from 0 (definitely will not) to 10 (definitely will). Persistence intentions scale scores were derived as the average of the items, with higher scores indicating stronger intentions. Prior validation research indicates that this measure, and similar measures of persistence intentions, are related to STEM persistence behaviors (e.g., applications to STEM-related graduate programs, post baccalaureate STEM career choices) [62, 82]. In addition, the current study found evidence of acceptable psychometric properties (e.g., longitudinal invariance and reliability), see supplemental materials.

**Scientific self-efficacy.** This six-item scale measured participants' confidence in their ability to complete a variety of scientific tasks (e.g., "Use technical science skills [use of tools, instruments, and/or techniques]") [46]. Participants rated their confidence on a scale from 1 (not at all confident) to 5 (absolutely confident). Self-efficacy scale scores were derived as the average of the six items, with higher scores indicating higher scientific self-efficacy. Prior validation evidence indicates that this measure of self-efficacy, as well as other similar measures of self-efficacy, are related to both mentoring supports and persistence in STEM [13, 46, 63]. In addition, the current study found evidence of acceptable psychometric properties (e.g., longitudinal invariance and reliability), see supplemental materials.

**Scientific identity.** This five-item scale measured the degree to which participants think of themselves as a scientist (e.g., "I have come to think of myself as a scientist.") [46]. Participants rated agreement with each statement on a scale from 1 (strongly disagree) to 5 (strongly agree). Identity scale scores were derived as the average of the five items, with higher scores

indicating higher scientific identity. Prior validation evidence indicates that this measure of identity, as well as other measures of identity, are related to both mentoring supports and persistence in STEM [13, 46, 63]. In addition, the current study found evidence of acceptable psychometric properties (e.g., longitudinal invariance and reliability), see supplemental materials.

**Internalization of scientific community values.** This four-item scale measured level of internalization of scientific community values [62]. Participants read descriptions of a person and rated "how much the person in the description is like you" (e.g., "A person who thinks discussing new theories and ideas between scientists is important."). Participants rated the degree to which each statement was like themselves on a scale from 1 (not at all like me) to 6 (very much like me). Values scale scores were derived as the average of the four items, with higher scores indicating higher internalization of scientific community values. Prior validation evidence indicates that this measure of identity is related to both mentoring supports and persistence in STEM [63, 65]. In addition, the current study found evidence of acceptable psychometric properties (e.g., longitudinal invariance and reliability), see supplemental materials.

**Mentor network diversity index.** Participants were instructed to think of a mentor as someone "who provides guidance, assistance, and encouragement on professional and academic issues. A mentor is more than an academic advisor and is someone you turn to for guidance and assistance beyond selecting classes or meeting academic requirements" [50]. On the pre-college survey, participants indicated (yes = 1 or no = 0) if people in any of the following roles served as a mentor in their lives (a) teachers, (b) guidance counselors, (c) program staff members, (d) graduate students, (e) peers, (f) professionals outside of the school setting, (g) family members/relatives, or (h) others. On the college surveys, a similar set of questions was administered; however, these questions focused on potential mentors within the college context. Specifically, participants indicated if people in any of the following roles served as a mentor in their lives (a) university faculty, (b) guidance counselors, (c) program staff members, (d) graduate students, or (e) peers. The mentor network diversity index scores were derived as the sum of the items, with higher scores indicating larger and more diverse mentor networks. Prior validation evidence indicates that measures of mentor network are related to developing deeper interest in STEM [50].

**Global measure of faculty mentoring practices.** Only students who indicated that they had a university faculty mentor were asked follow-up mentoring practices questions about their faculty mentor. This 16-item scale measured the degree to which the faculty mentor provided instrumental and psychosocial support in a global fashion over the last six months (e.g., "To what extent has your mentor conveyed feelings of respect for you?") [83, 84]. Participants rated the degree of support received on a scale from 1 (not at all) to 5 (to a large extent). Faculty mentoring support scale scores were derived as the average of the 16 items, with higher scores indicating higher levels of mentoring support. This scale was administered on four occasions from the first through the fourth years of college (i.e., T2-T5). Prior validation evidence indicates that this measure of the mentoring practices is related to the development of scientific identity, science values, and persistence-related activities, such as scholarly productivity [63, 65, 84]. In addition, the current study found evidence of acceptable psychometric properties (i.e., reliability), see supplemental materials.

**Research experiences index.** This 10-item index asked participants to report on their level of involvement in research experiences [85]. Participants indicated their involvement (yes = 1 or no = 0) in a variety of research activities over the prior year (e.g., hands-on research in a class, research in a laboratory or on a research team outside of class, presented original research at a national or regional conference). Research experience index scores were derived as the sum of the items, with higher scores indicating more research experiences. This checklist

was administered on four occasions from the first through the fourth years of college (i.e., T2-T5). Prior validation evidence indicates that this, and other measures of research engagement, are related to the development of scientific self-efficacy, identity, values, and persistence in STEM [63, 85, 86].

## Plan of analysis and preliminary analyses

**Assessment of model fit in SEM.**  All longitudinal analyses were conducted in a structural equation modeling (SEM) framework using *Mplus* version 8.00 [87]. The adequacy of model-data fit in SEM analyses was assessed with a variety of global fit indices, such as the $\chi^2$ test, root-mean-square error of approximation (*RMSEA*), comparative fit index (*CFI*), and standardized root-mean-square residual (*SRMR*) [88–91]. The observed fit index values compared to values representing acceptable model fit, such as CFI $\geq$ .95, RMSEA $\leq$ .05 (or a 90% CI that included .05 but did not include .10), or SRMR $\leq$ .08. In addition, when nested models were compared, the cutoff value of $\Delta CFI$ values $\geq$ .01 or $\Delta RMSEA$ values $\leq$ -.015 indicated worse model fit [92].

Consistent with recommendations for controlling Type-I error rate inflation in complex or exploratory SEMs, we used the Benjamini-Hochberg False Discovery Rate (FDR) procedure when evaluating the statistical significance of structural coefficients in the final substantive model (i.e., Model-8) [93–95].

**Statistical assumption checking.**  Prior to substantive analyses, two sets of preliminary analyses were conducted. First, we examined the patterns of missing data, screened for outliers, and tested the statistical assumptions related to distributions. Response rates to the survey varied across occasions (Table 1). The response rate among the 751 STEM students in this study was 82.3% at T1, 68.7% at T2, 81.8% at T3, 81.6% at T4 and 73.1% at T5. Furthermore, 33.1% of STEM students completed all five surveys (i.e., 100% completion rate across all waves of data collection), 25.2% of the STEM students completed four out of five surveys (i.e., 80% completion rate), 15.5% of the STEM students completed three out of five surveys (i.e., 60% completion rate), 12.6% of the STEM students completed two out of five surveys (i.e., 40% completion rate), and 13.7% of the STEM students completed only one out of five surveys (i.e., 20% completion rate). Little's MCAR test [96] revealed that the data were not missing completely at random $\chi^2(6,370) = 6,860.51$, $p < .001$. Therefore, full information maximum likelihood (FIML) estimation methods were used, as FIML estimation has been shown to ensure unbiased estimates under the more reasonable missing-at-random (MAR), even when the percent of missing data is large [97]. That is, MAR allows missing data when they are conditioned on observed data used in the analysis [98]. Recent research has argued that longitudinal panel studies, such as this one, are highly unlikely to violate the MAR assumption because missing data on each variable can be conditioned on the data from the same variable collected on prior waves [99]. This approach has been shown to be adequately address missing data in large longitudinal studies, such as the current investigation [100]. In addition to missing data analyses, outlier analyses (using leverage values, Studentized deleted residuals, and Cook's distance values) and distributional assumptions (i.e., normality of residuals, homoscedasticity of residuals, and linearity) were conducted [101]. The analyses revealed no extreme outliers and confirmed the tenability of distributional assumptions.

**Longitudinal and cross-group measurement invariance CFA models.**  The second set of preliminary analyses focused on the assumption of longitudinal measurement invariance for our outcome and social influence process (mediating) variables [89]. Unfortunately, it was not possible to estimate the measurement invariance of the global measure of faculty mentoring practices in the present study due to the relatively small sample size of college students that

reported having a mentor at any given time point across their college tenure. That is, the number of parameter estimates for CFA models of the global measure of faculty mentoring practices vastly exceeded the number of participants with a faculty mentor, which resulted in models that did not converge. For all other latent constructs of interest consistent with best practices, individual tests of longitudinal measurement invariance for the outcome and mediators were conducted separately for men and women [89]. Furthermore, we tested multiple-groups measurement invariance (men, women) for the outcome and mediators at each time point. Maximum likelihood estimation with robust standard errors were used in CFA models to account for non-normality in the item-level data, while ML estimation was used with scale scores which were more consistent with distributional assumptions.

Consistent with expectations, the results of the CFA models indicated acceptable model-data fit for tests of longitudinal measurement invariance within groups (S2 Table). Furthermore, tests of cross-group (men, women) measurement invariance indicated acceptable data-model fit at each time point (S2 Table), with the exception that model fit was slightly below the cutoff levels at T2 (i.e., first-year of college). An examination of the standardized residual covariances indicated stronger than typical relationships between scientific persistence intentions (i.e., indicators-1, intention to pursue a scientific research career) with science values (i.e., indicators-4, "scientific research can solve many of today's world challenges") for both men and women in the first-year of college. This pattern did not repeat at any other time-points and thus we proceeded with analysis. As a final assessment of cross-group invariance, we compared the fit of models that held correlations among latent factors to be invariant for men and women at each time point. Fit indices indicated that model fit was not worsened by constraining correlations to be equivalent across groups (S2 Table). Therefore, the gender groups were combined for all substantive analyses.

## Results

Prior to formally testing mediation models in SEM, as noted above, we tested and confirmed (a) longitudinal measurement invariance, (b) cross-gender measurement invariance, and (c) cross-gender invariance in the pattern and magnitude of correlations among constructs at each time-point. Having found no evidence of moderation due to gender status, the gender was not considered further in any of the statistical analyses. Next, we examined the descriptive statistics and pattern of correlations among the social influence factors, social influence processes, and social integration (i.e., persistence intentions) at each time point (Table 1 and S1 Table). Consistent with expectations from TIMSI, faculty mentorship, mentor network diversity, and research experiences exhibited small-to-moderate positive correlations (i.e., $r$'s .10-.30) with science efficacy, science identity, science community values, and scientific career persistence intentions over time (S1 Table). Furthermore, science efficacy, science identity, and science community values exhibited moderate positive correlations with scientific career persistence intentions (i.e., $r$'s .25-.60; S1 Table).

### Longitudinal SEMs

Having found correlations that were consistent with our expectations, we next conducted a set of preliminary analyses concerned with identifying the most parsimonious and best fitting longitudinal SEM (see supplemental materials for a complete discussion of Models 1–8; S1 and S2 Figs). As shown in Fig 2, the series of tests of eight nested SEMs revealed that the longitudinal model that provided maximum parsimony without sacrificing good model-data fit included (a) freely estimated first- and higher-order relations within each construct (i.e., stability paths), (b) first-order cross-lagged relationships across constructs (i.e., cross-lagged paths), (c)

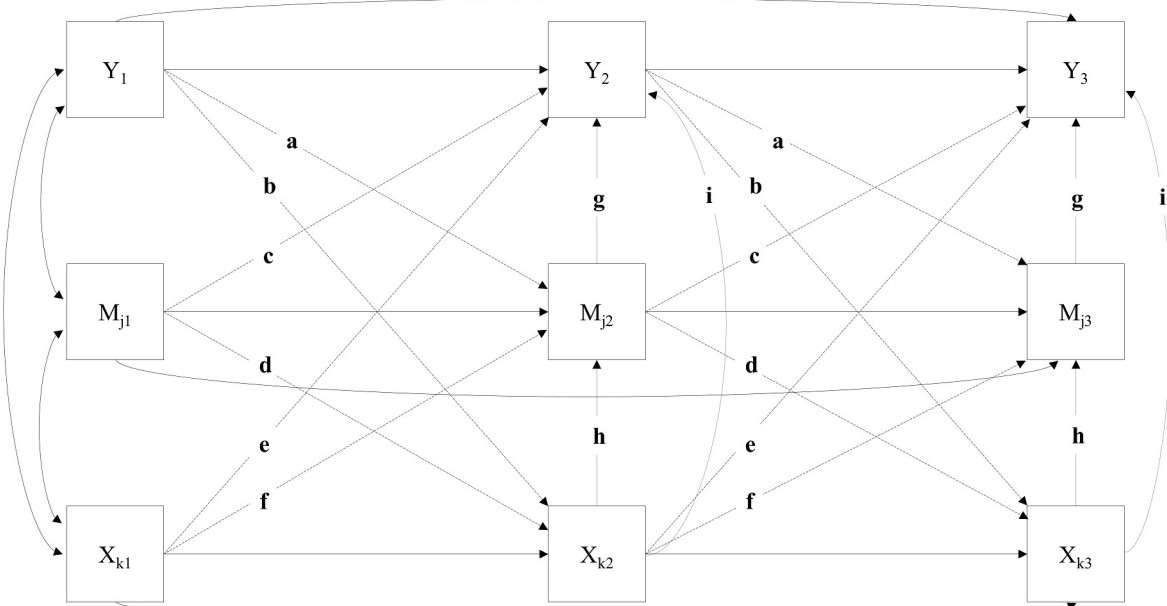

**Fig 2. Three time point conceptual representation of the final simplified longitudinal model with cross-lagged and contemporaneous equilibrium constraints (Model-8).** Y = outcome (i.e., scientific career persistence intentions [indicator of integration]), $M_j$ = mediators (i.e., science self-efficacy, science identity, and science community values [indicators of social influence processes]), $X_k$ = contextual factors (i.e., faculty mentor support, mentor network diversity, research experiences [indicators of social influence agents]). Stability paths shown as solid lines, first-order longitudinal cross-lagged paths shown as dashed lines, and contemporaneous mediation paths from the contextual factors to the outcome are shown as dotted lines. Subscripts $_1$ = T1 spring/summer prior to college, $_2$ = T2 spring first year of college, $_3$ = T3 spring second year of college, but T4 fall third year of college, and T5 spring fourth year of college are not shown for the sake of parsimony.

relationships across constructs within each time point (i.e., contemporaneous paths), (d) cross-lagged developmental equilibrium constraints (holding cross-lagged paths equal across time), and (e) contemporaneous equilibrium constraints (holding contemporaneous paths equal across time; see Fig 2; for complete details of nested model comparisons see the supplemental materials and S3 Table). For example, Fig 2 shows cross-lagged developmental equilibrium in that paths from the influence agents (i.e., Xs) to the social influence processes (i.e., Ms) are constant over time (i.e., paths are denoted with the symbol "f" to show that these coefficients held constant from T1→T2, T2→T3, etc...). Similar notation was used in Fig 2 to denote all aspects of cross-lagged developmental (i.e., paths labeled a-f), as well as, contemporaneous equilibrium (i.e., paths labeled g-i).

## Social influence operates contemporaneously and reciprocal influence operates longitudinally

Next, we inspected the parameter estimates from Model-8 to address research questions concerning the mediated pathways linking the social influence agents to persistence intentions, as well as reciprocal influence. The contemporaneous paths linking social influence agents to persistence intentions through the social influence process were consistent with our expectations (paths coefficients are represented in the lower section of Table 2 and are also shown in diagram form in S3 Fig. as paths under the "T5 (4th Year of College);" S3 Fig. shows a graphical representation of the statistically significant standardized coefficients in Table 2). More specifically, mentor network diversity, faculty mentor support, and research experiences exhibited small-to-moderate positive predictive relationships with science self-efficacy and science

**Table 2. Trimmed summary of the standardized coefficient from the final model highlighting reciprocal social influence from the third to fourth year of college (Model 8, *N* = 751).**

| Time | Predictors | Outcome | Social Influence Processes | | | Social Influence Agents | | |
|---|---|---|---|---|---|---|---|---|
| | | Persistence Intentions | Science Efficacy | Science Identity | Science Values | Mentor Network Diversity | Faculty Mentor Support | Research Experiences |
| T4 3rd year of college | Persistence Intentions[a] | .36*** | .07** | .13*** | .08*** | .03 | -.02 | .07*** |
| | Science Efficacy[b] | -.08*** | .26*** | | | -.02 | .06 | -.03 |
| | Science Identity[b] | -.13*** | | .23*** | | .05 | -.05 | .10*** |
| | Science Values[b] | -.04* | | | .11 | -.01 | .04 | .01 |
| | Mentor Network Diversity[c] | .02 | -.06* | -.03 | -.04 | .36*** | | |
| | Faculty Mentor Support[c] | .05 | -.03 | -.03 | -.05 | | .35*** | |
| | Research Experiences[c] | .004 | -.02 | -.06 | -.03 | | | .46*** |
| T5 4th year of college (Contemporaneous) | Science Efficacy[b] | .05* | | | | | | |
| | Science Identity[b] | .33*** | | | | | | |
| | Science Values[b] | .17*** | | | | | | |
| | Mentor Network Diversity[c] | .01 | .07** | .06** | .06** | | | |
| | Faculty Mentor Support[c] | -.01 | .15*** | .12*** | .12** | | | |
| | Research Experiences[c] | .05 | .08** | .12*** | .06 | | | |
| $R^2$ | | .58 | .34 | .43 | .21 | .28 | .20 | .37 |

a = outcome

b = social influence processes

c = social influence agents. T4 = fall 3rd year of college, T5 = spring 4th year of college. All standardized structural coefficients ascertained from STDXY in Mplus as all variables were continuous. Underlined values represent stability coefficients; coefficients in standard text associated with predictors from the prior year in college are first-order cross-lagged coefficients, and coefficients associated with predictors from the current year of college are contemporaneous. The B-H FDR procedure was used to determine the statistical significance of all unstandardized coefficients. Based on the FDR procedure, all *p*-values less than .023 are reported statistically significant.

*$p \leq .023$

**$p \leq .01$

***$p \leq .001$.

identity. For example, the standardized path coefficient linking faculty mentor support with science identity was in the small-to-moderate range (i.e., $\beta$ = .12) and the standardized coefficient linking mentor network diversity to science identity was very small (i.e., $\beta$ = .06). Furthermore, mentor network diversity and faculty mentor support showed small-to-moderate positively predictive relationships with science community values (Table 2). In general, the quality of faculty mentor support was the strongest predictor, as evidenced by having relatively larger standardized coefficients. Regarding the information presented in Table 2, given that (*a*) our research questions concern cross-lagged and contemporaneous paths and (*b*) Model-8 constrains cross-lagged and contemporaneous coefficients to be invariant across time, Table 2 presents only one example time-lag (third to fourth year of college). The relevant coefficients do not change across time and thus one time-lag is representative of all other time-lags. However, all time-lags (High school through fourth year of college) are shown in S1 and S4 Tables.

Next we examined the paths linking the social influence processes (i.e., science self-efficacy, science identity, and science community values) to contemporaneously measured persistence intentions and found that they exhibited small-to-moderate positive predictive relationships with scientific career persistence intentions (Table 2, paths coefficients are represented in the lower section). For example, the standardized path coefficient linking science identity with persistence intentions was moderate in size (i.e., $\beta = .33$) and the standardized coefficient linking science efficacy to persistence intentions was very small (i.e., $\beta = .05$). In general, science identity was the strongest predictor, followed by science values, followed by science efficacy. Table 2, which only shows the T3 → T4, is used for illustration purposes because the effects were constrained to be consistent over time in the cross-lagged model.

Next we examined the longitudinal paths linking prior engagement with influence agents with future levels of social influence processes and social integration, as well as longitudinal reciprocal relationships. Inconsistent with expectations, the longitudinal cross-lagged paths linking prior engagement with social influence agents to future persistence intentions through the social influence processes were all negative in sign (Table 2). By contrast, bivariate correlations of cross-lagged associations were positive in sign (S1 Table). For example, the standardized path coefficient linking prior science identity with future persistence intentions was small-to-moderate and negative (i.e., $\beta = -.13$); however, the bivariate correlations between prior science identity and future persistence intentions were moderate and positive (i.e., $r$s range from .41-.43, see S1 Table). Thus, we interpret the negative sign of the cross-lagged coefficients under Model-8 as indicating a "classical" suppression effect due to multiple mediators and contemporaneous mediated effects [74, 102]. Therefore, engagement with social influence agents appears to be operating through contemporaneous, rather than cross-lagged processes. For example, T4 mentoring and research experiences only influence T5 science efficacy, identity, values, and persistence intentions through the effects on their T4 counterparts (i.e., through their respective stability paths).

Next, we examined the reciprocal paths linking prior persistence intentions to future engagement with social influence agents through the social influence processes. The results indicated that (a) prior scientific persistence intentions exhibited small-to-moderate positive influence on future science efficacy, identity, community values, and engagement in research experiences, and (b) prior science identity exhibited small-to-moderate positive influence on future research experiences (Table 2; presented in the top of the table and also shown in diagram form in S3 Fig. as paths linking "T4 (3rd Year of College)" to "T5 (4th Year of College)"). For example, the standardized path coefficient linking prior science career persistence intentions with future science identity was small-to-moderate (i.e., $\beta = .13$), the path linking prior science career persistence intentions with future science efficacy was very small (i.e., $\beta = .07$), and the path linking prior science identity with engagement in undergraduate research was small (i.e., $\beta = .10$). Therefore, early levels of social integration (scientific persistence intentions) exhibit reciprocal direct and indirect effects on future engagement with social influence agents.

## Mediation

A bootstrapping procedure with 20,000 repetitions was used to estimate percentile-based confidence intervals around the contemporaneous and reciprocal mediated effects [103]. Since the contemporaneous coefficients did not vary over time due to contemporaneous equilibrium, we report the mediated effects for a single time point (i.e., T5 or fourth year of college; Fig 3). As shown in Fig 3, the analysis revealed small, positive, statistically significant contemporaneous mediated effects for all of the social influence agents on persistence intentions through

## Contemporaneous Mediation

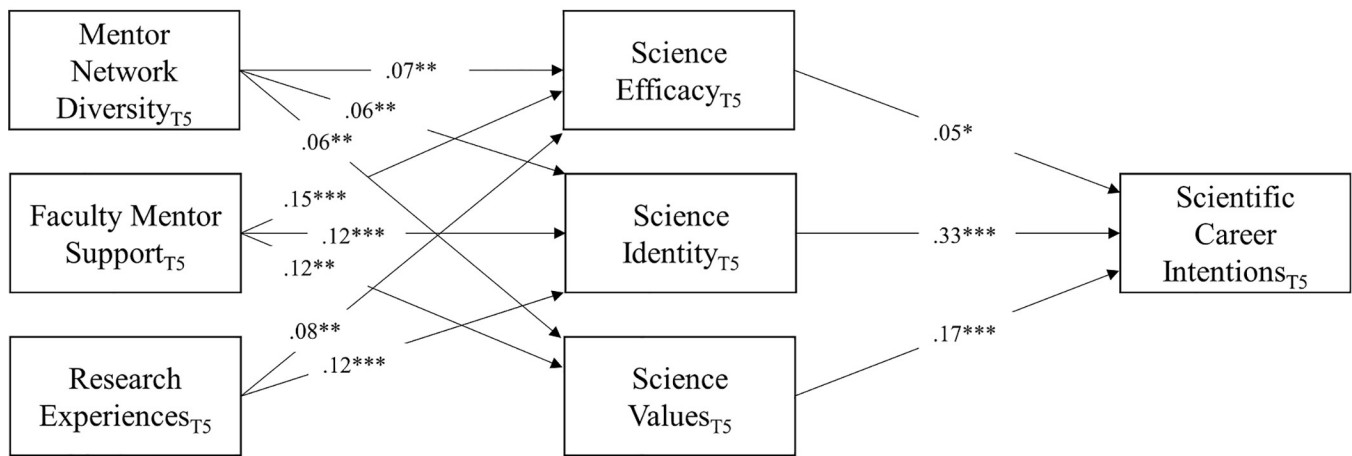

Mentor Network Diversity to Intentions through Self-Efficacy: $\beta_{a\times b}$ = .004; $a\times b$ = 0.004, percentile bootstrapped 95% *CI* [0.000, 0.011]
Mentor Network Diversity to Intentions through Identity: $\beta_{a\times b}$ = .02; $a\times b$ = 0.03, percentile bootstrapped 95% *CI* [0.01, 0.05]
Mentor Network Diversity to Intentions through Values: $\beta_{a\times b}$ = .010; $a\times b$ = 0.014, percentile bootstrapped 95% *CI* [0.004, 0.025]

Faculty Mentor Support to Intentions through Self-Efficacy: $\beta_{a\times b}$ = .008; $a\times b$ = -0.019, percentile bootstrapped 95% *CI* [-0.001, 0.049]
Faculty Mentor Support to Intentions through Identity: $\beta_{a\times b}$ = .04; $a\times b$ = 0.12, percentile bootstrapped 95% *CI* [0.05, 0.19]
Faculty Mentor Support to Intentions through Values: $\beta_{a\times b}$ = .02; $a\times b$ = 0.06, percentile bootstrapped 95% *CI* [0.02, 0.10]

Research Experiences to Intentions through Self-Efficacy: $\beta_{a\times b}$ = .004; $a\times b$ = 0.004, percentile bootstrapped 95% *CI* [0.000, 0.010]
Research Experiences to Intentions through Identity: $\beta_{a\times b}$ = .04; $a\times b$ = 0.04, percentile bootstrapped 95% *CI* [0.03, 0.06]

**Fig 3. Contemporaneous mediated paths linking social influence agents to intentions through the social influence processes at T5 (Model-8).** T5 = spring 4[th] year of college. Only one model is shown since the contemporaneous mediation was invariant across the first, second, and third years of college. The full SEM is not shown for the sake of parsimony. All coefficients are standardized (STDXY). Confidence intervals were estimated using percentile bootstrapping method. $^*p \leq .023$, $^{**}p \leq .01$ $^{***}p \leq .001$.

science identity and/or science community values. More specifically, we found that faculty mentor support exhibited the largest indirect effect on scientific career persistence intentions through science identity (i.e., $\beta_{a\times b}$ = .04), followed by research experiences and mentor network diversity through identity. Relatively smaller positive indirect effects were observed for indirect effects through scientific community values.

Finally, we examined reciprocal longitudinal mediation. Since the cross-lagged coefficients did not vary over time due to developmental equilibrium, we report the mediated effects for a single two-year time-lag (i.e., T3 →T4 →T5 or 2[nd] through 4[th] years of college). As shown in Fig 4, the analysis revealed small, positive, longitudinal reciprocal mediated effects of T3 scientific persistence intentions on T5 engagement in research experiences through T4 science identity.

## Discussion

### Social influence leads to short-term student integration

The TIMSI framework posits that persons and social contexts can operate as social influence agents. Influence agents integrate the targets of social influence into a social system by rewarding compliance with social norms (or sanctioning non-compliance), cuing the role by which the target to influence is identified, and/or reinforcing how the groups' values are consistent

## Longitudinal Reciprocal Mediation

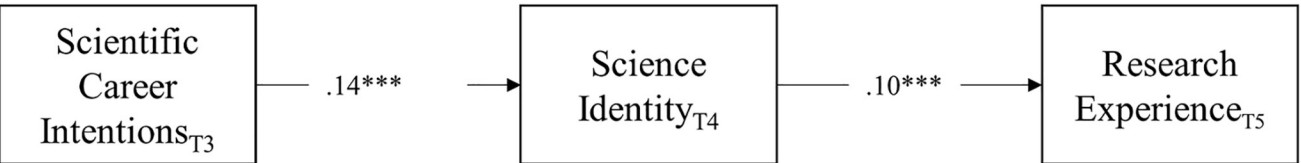

Intentions to Research Experiences through Identity: $\beta_{a \times b}$ = .014, $a \times b = 0.013$, percentile bootstrapped 95% *CI* [0.006, 0.021]

**Fig 4. Reciprocal longitudinal mediated paths linking early intentions to later engagement with influence agents through science identity (Model-8).** All coefficients are standardized (STDXY). The full SEM is not shown for the sake of parsimony. Confidence intervals were estimated using percentile bootstrapping method. Subscripts T3 = spring second year of college 2017, T4 = fall third year of college 2017, T5 = spring fourth year of college 2018. *$p \leq$.023, **$p \leq$.01 ***$p \leq$.001.

with the target of influence's internalized values. The result of successful influence is greater efficacy to engage in normative behaviors, stronger identification with the group of the influencing agent, and internalization of that group's values. Partially consistent with TIMSI predictions (Fig 1 solid lines Social Influence Agents → Social Influence Processes → Social Integration), we found that among students from HO group in STEM higher levels of engagement with influencing agents of the scientific academic community (operationalized as quality of faculty mentor support, mentor network diversity, and undergraduate research experiences) predicted higher levels of social influence processes (operationalized as science self-efficacy, identity, and internalization of community values). Science efficacy, identity, and values, in turn, predicted higher levels of social integration (operationalized as persistence intentions). Furthermore, although the pattern of predictive associations was consistent with those hypothesized from TIMSI, not all variables were found to be equally predictive (i.e., mentor network diversity and science efficacy exhibited very small effects).

These findings are consistent with prior research showing moderate positive associations between the quality of faculty mentorship and psychosocial outcomes, but weaker positive associations with persistence-related outcomes [45–47, 63, 104]. Taken together, the current and previous findings indicate that faculty mentors exert influence and draw their mentees into their disciplinary community by helping them to see themselves as belonging in the community, by internalizing community values, and to a lesser extent, seeing themselves as competent. The present findings reaffirm the unique and influential role that faculty mentors can play in drawing students into the STEM community. However, high quality faculty mentorship can be a relatively scarce commodity in higher education. Accordingly, our parallel findings concerning research experiences suggest potentially more scalable ways to support students interested in STEM careers.

The present findings are partially consistent with theory and empirical evidence on the direct and indirect impacts of undergraduate research experiences on integration into the scientific community [52, 56, 62, 63, 85, 105–110]. That is, our study found that research experiences indirectly support persistence through the development of science identity and internalization of community values; however, these indirect effects only manifested contemporaneously (i.e., within an academic year, not longitudinally). The present findings reinforce

calls to broaden access to undergraduate research experiences through high-quality and institution-wide scalable solutions, such as course-based undergraduate research experiences or CUREs [14, 52, 111, 112]. That is, recent work has shown that CUREs (a.k.a., CREs) can be a cost-effective and scalable way to engage large numbers of undergraduates in authentic and transformative research experiences at any-point during the undergraduate tenure [52, 113, 114].

Mentor network diversity, a novel indicator of quantity of interactions with social influence agents, proved to be the weakest predictor of social influence. However, we suspect that refinements to the operationalization of mentor network characteristics may produce stronger effects than were shown in the present study. Specifically, recent research has shown unique benefits associated with having peer mentors, as well as being involved in an undergraduate–postgraduate–faculty triadic mentoring relationship [49, 66, 67, 115]. Therefore, future research on the effects of mentor networks may need to measure specific network structures and the quality of those relationships, rather than mentor network role diversity, as was done in the present study. While the unique effects of mentor network diversity predicting integration were not robust in the present study, the overall results suggest that curricular and co-curricular efforts to promote diversified peer- and community-based approaches to help students develop their network of mentors may be beneficial to integration [13, 50, 55, 68, 72, 105, 110, 115–117].

Finally, it was notable that science identity and values were the two primary mechanism through which social influence agents (i.e., faculty mentors and research experiences) impacted contemporaneous persistence intentions. That is, although science self-efficacy was moderately correlated with persistence intentions at each time point (see S1 Table), self-efficacy was a relatively weak unique predictor in the longitudinal models. The TIMSI framework posits that the motivations to comply with the rules, roles, or values of influence agents may differ across individuals [61] and we propose that it may also differ across developmental stages. That is, we suspect that this pattern of results points to a developmental process, whereby self-efficacy, identity, and values impact integration and persistence differently, depending upon professional career stage. Specifically, science self-efficacy (i.e., feeling confident one can do the science) may be the most important process in supporting social integration for students at earlier points in the career development continuum (i.e., pre-college). However, for undergraduates, self-efficacy may be a necessary, but not the unique or sufficient determining orientation that predicts persistence, with science identity becoming a stronger predictor. Data drawn from graduate students and postdoctoral fellows finds that values become a strong predictor when scholars are choosing to stay or leave academia [118, 119]. Future research across a longer developmental span (e.g., secondary school–college–graduate school and beyond) are needed to continue to elaborate on and test this developmental hypothesis.

Although some aspects of the findings converge with the prior literature, the longitudinal analyses also revealed that the social influence process leading from agents to integration only operated contemporaneously. That is, the effects of influence agents on social integration into the scientific community was fleeting, in that the impact was present when the agent was present, but the impact disappeared a year later when the influence agent was removed or withdrew. This finding is consistent with classical social influence literature showing that proximity, which can be physical or temporal, to an influence agent determines the strength and longevity of the social influence [120, 121]. Kelman [59] noted in his early writing that "surveillance by influencing agent," "salience of relationship to agent," and "relevance of values to issue" each made social influence more likely to occur (p. 67). Proximity in time and space

to a mentor or engagement in a socializing research experience would make each of these more likely, resulting in greater influence occurring.

**Longitudinal findings depart from earlier longitudinal research.** The absence of a longitudinal mediated effect of interactions with influence agents on social integration was unexpected and departed from earlier research findings. Specifically, previous research had shown that mentorship quality or undergraduate research experiences influenced later science self-efficacy, identity, values, and/or integration into the STEM communities and careers [45, 46, 62, 63, 65]. We did not find a similar pattern in the present study. Several factors may be at play in distinguishing the current finding of a fleeting relationship to previous findings of a longitudinal relationship. Most notably, methodological differences may help to explain the departure of the current pattern of findings from patterns found prior research. The current and prior studies operated under different protocols for the timing of repeated measurements. For example, one study followed undergraduates over relatively shorter intervals of time and collected fewer observations over time during students' undergraduate tenure (e.g., three times over a 10-week summer research experience) [65], while another longitudinal study tracked students on an irregular basis (e.g., junior year, senior year, and 4-years post baccalaureate attainment) [63]. By contrast, the present study tracked students annually from the senior year of high school through their senior year of college. The optimal timing for repeated measurements to fully capture the influence process is not yet well understood; however, based on the present and prior findings, it appears that socializing agents may exert more rapid impact on integration than could be fully captured by annual assessments. Influence may be bounded by the beginning, middle, and ending of a socializing experience (e.g., one- or two-semester course-based research experiences). Future research should attend to the natural timing of socializing events to fully capture their potential longitudinal influence. In addition, the current and prior studies utilized different approaches to modeling typical and reciprocal influence. Prior studies have not examined reciprocal influence, whereas the current study was the first to heed recommendations for best practices for longitudinal mediation by utilizing a full forward model that allowed for reciprocal influence when testing the TIMSI predictions [74, 75, 89]. It is possible that if prior studies had used a similar approach, that similar pattern may have emerged.

Another possible factor contributing to the different patterns of longitudinal results concerns demographics. That is, much of the prior research has focused on the social integration of HU students into STEM careers and furthermore, one of the few prior longitudinal studies that tested TIMSI predictions did so among undergraduates from HU groups in STEM [63]. Estrada and colleagues [63] found that science identity alone mediated the relationship between agents and integration (operationalized as post-baccalaureate persistence in a STEM field). By contrast, the present study, which focused on undergraduates from HO groups in STEM, found that both identity and values mediated the relationship between agents and integration. These differences may indicate that undergraduates from under- and over-represented groups are socializing into the scientific community via distinct social influence processes. Previous research on workforce development has shown that even when socioeconomic and academic preparation were controlled, undergraduates from HU groups are not persisting (and, in turn, integrating) into their fields of study at the same rate as students from HO groups, and this difference can be attributed to how they are socially experiencing their academic environment [122]. Thus, there may be culturally different emphases on what psychosocial variables are most central to the integration process for undergraduate students. The integration process is further complicated by a potential intersection with developmental processes. That is, culturally different emphases may change at different developmental periods, as there is emerging evidence that values play an increasingly important role in STEM career

decision making among graduate students and postdocs from HU groups [119]. However, future research measuring the emergent patterns across the undergraduate tenure could identify true subgroup similarities or differences in the patterns of social integration into STEM fields.

## Integration leads to engagement with social influence agents

Consistent with TIMSI predictions, our study found compelling evidence of longitudinal reciprocal influence into the scientific community via two distinct processes. First, students with stronger intentions to persist in the scientific community at an earlier time point were more likely to seek out and engage in research experiences one year later. This finding is consistent with social psychological theories of human behavior (e.g., SCCT) that hypothesize a direct reciprocal link from performance or persistence outcomes to future engagement with learning experiences that support the development of continued motivation [40, 41, 44]. Second, students with stronger persistence intentions at an earlier time point developed higher levels of science self-efficacy, identity, and values one year later, which, in turn, promoted higher levels of research engagement the following year. This finding is consistent with TIMSI, from which we expected that the reciprocal link from integration to interactions with influencing agents of the scientific community would be mediated by social influence processes (see dashed lines in Fig 1, Social Integration → Social Influence Processes → Social Influence Agents).

Several potential implications of the robust evidence for reciprocal influence should be noted. One is that individuals, particularly individuals from HO groups, are actively engaged in self-socialization–seeking out relationships and experiences that cyclically reinforce their integration into their disciplinary community. This finding affirms hypotheses of reciprocal or cyclical influence as described in multiple social psychological theories of human behavior [20, 40, 41, 61]. However, these findings also put meat on the bones of heretofore generic reciprocal hypotheses by showing that early levels of social integration have both direct influence on future engagement with socializing experiences, as well as indirect influence of future engagement with socializing experiences through the development of stronger identity. Therefore, social influence and other motivational theories may need to be revised regarding the thinking on reciprocal influence to include both direct and indirect pathways.

These findings also suggest that future research studies should identify "if" and "when" HU students are as equally likely as HO students to seek out socializing activities. Given the present and prior findings, it is an open question as to whether or not a similar reciprocal cycle may be occurring for HU students. Future studies could examine if HO and HU students experiencing greater science efficacy, identity, and value endorsement take similar initiative to seek out research experiences. Contextualizing the findings by assessing such experiences as microaggressions and microaffirmations from influencing agents could also inform interpretation of the outcomes.

## The TIMSI socialization process is highly stable in the college years

The present study addressed a gap in the literature concerning the degree to which TIMSI predictions varied at critical time points in the undergraduate tenure from the senior year of high school through the senior year of college. For example, a lingering question in the research experiences literature has concerned relative impact of early versus later exposure to undergraduate research experiences [85]. Our study found no strong evidence for a critical time point for social influence. Rather, formal tests of developmental and contemporaneous equilibrium indicated a high degree of consistency in the associations among mentoring and research

experiences, social influence processes, and social integration from the first through the fourth years of college. In addition, the present study found evidence of a high degree of consistency in the measurement of the social influence and social integration constructs over time and across genders. More specifically, formal tests of measurement invariance revealed that (a) each social influence and integration construct exhibited stability of measurement (in terms of factor loadings) from high school through senior year of college (longitudinal measurement invariance), (b) the measurement of each construct operated equally well for men and women at each time point (multiple groups measurement invariance), and (c) the strength and direction of associations among the constructs were equivalent for men and women at each time point (multiple groups correlation invariance).

## Limitations and implications for future research

Although the present study provides unique insights into how HO students integrate into their disciplinary communities, several notable caveats limit generalizations of the findings. The present study made use of a cohort of HO students pursuing a STEM major at a single large northeastern public research-intensive university. As described above, the pattern of associations linking interactions with influence agents to social integration may operate differently for students from HU groups or less research intensive colleges or universities. Therefore, additional research across diverse types of institutions and diverse samples of HU and HO student groups pursuing STEM degrees using the same rigorous modeling approach will be needed to show for whom and under what conditions the observed patterns hold. For example, it would be informative to test for potential subgroup differences in the degree to which influence operates contemporaneously or longitudinally. Moreover, further study is needed on the degree to which TIMSI derived hypotheses hold for first- versus multi-generational students in STEM fields when accounting for the influence of pre-college factors [123]. In addition, the present study used a single and self-reported operationalization of social integration. Although the present measure of social integration has been shown to be predictive of post-baccalaureate STEM career choices among HU students [63, 82], diverse measures of social integration, such as major, course taking, belonging to pre-professional societies and/or national organizations, applications to graduate school, and post-graduation employment would broaden the results and determine the limits of these findings. Furthermore, diverse and objective or non-self-reported measures of social integration would reduce the potential threat that common method bias may have had on the magnitude of associations between predictors and integration in the present study. Similarly, the present study focused on mentee perceptions of the quality of psychosocial and instrumental support received from a faculty mentor, rather than other aspects of mentoring relationships (e.g., relationship duration, the mentor's perspective). Future research should ascertain reports from both mentor and mentee perspectives to gain insights into the influence of mentoring relationships on social influence processes and social integration. Furthermore, as noted above, the current study measured all constructs annually, which may not have been sufficiently frequent to fully characterize more rapidly mediated links from agents of the scientific community to social integration, through the social influence processes. Therefore, future research will need to consider timing–and, particularly more rapid timing of measurement to illuminate the short-term longitudinal social influence paths.

The sum of the distinctions between the present and prior research endeavors likely accounts for much of the variation in findings and points to recommendations for future research. Future longitudinal research on TIMSI predictions should attend to (a) the possibility of distinct socialization processes for students from HU and HO groups in STEM, (b) examining the entire undergraduate tenure, (c) the frequency and timing of repeated

measurement occasions, (d) capture a breadth of theoretically meaningful socializing experiences and all TIMSI identified social influence processes, (e) the inclusion of a variety of measures of integration into STEM career pathways, and (f) the close adherence to best practices in longitudinal mediation and causal ordering research.

## Conclusion

The overarching purpose of this longitudinal study was to improve our understanding of STEM workforce development by testing TIMSI predictions about the ways in which individuals socialize into STEM communities and careers. In particular, this study followed a cohort of HO students from high school through their senior year of college. We examined the process whereby agents of the scientific academic community exert influence on integration into the scientific community through social influence processes. Advancing research in this area, the findings also measure the reciprocal influencing processes whereby those who are most integrated seek out more exposure to additional influencing opportunities–resulting in a cycle of deeper integration. The results of this study clearly show that (a) agents of the scientific academic community exert contemporaneous or fleeting indirect influence on social integration, (b) early social integration exerts reciprocal longitudinal influence on future engagement with agents of the scientific academic community, and (c) the socialization process described by the TIMSI framework is highly stable from senior year of high school through senior year of college.

Practically, these results demonstrate that faculty mentors exert influence and draw their mentees into their disciplinary community by helping them to see themselves as belonging in the community and by helping them to internalize scientific community values. Further, students benefit from having increased integration into their professional communities by seeking out more opportunities to socialize into their communities. Therefore, faculty interested in broadening participation may need to support students who are not proactive in finding mentors or research opportunities through systematic curriculum-infused opportunities for authentic research engagement, such as course based undergraduate research experiences. In sum, this study shows how using the lens of social influence theory is useful for identifying processes that may improve interventions to increase academic persistence for all students and to support workforce development.

## Supporting information

**S1 Fig. Three time point conceptual full-forward plus contemporaneous mediation model.** Y = outcome (i.e., scientific career persistence intentions [indicator of integration]), $M_j$ = mediators (i.e., science self-efficacy, science identity, and science community values [indicators of social influence processes]), $X_k$ = contextual factors (i.e., faculty mentor support, mentor network diversity, research experiences [indicators of social influence agents]). Stability paths shown as solid lines, first- and higher-order longitudinal cross-lagged paths shown as dashed lines, and contemporaneous mediation paths from the contextual factors to the outcome are shown as dotted lines. Subscripts $_1$ = T1 spring/summer prior to college 2015, $_2$ = T2 spring first year of college 2016, $_3$ = T3 spring second year of college 2017, but T4 fall third year of college (2017) and T5 spring fourth year of college (2018) are not shown for the sake of parsimony.
(TIF)

**S2 Fig. Planned tests of simplifications of the full-forward plus contemporaneous mediation models.** Model-1 represents the full-forward (i.e., first- and higher-order stability and

cross-lagged paths) plus contemporaneous mediation model. Model-2 represents only first-order stability and cross-lagged plus contemporaneous mediation model. Model-3 represents first- and higher-order stability and first-order cross-lagged plus contemporaneous mediation model. Model-4 represents first- and higher-order cross-lagged and first-order stability plus contemporaneous mediation model. Model-5 represents the full forward without contemporaneous mediation model. Model-6 represents developmental equilibrium of the cross-lagged coefficients (i.e., invariance constraints) placed on the simplest and best fitting model identified from Models 1–5. Model-7 represents developmental equilibrium of the stability coefficients added to the simplest and best fitting model from Models 1–6. Model-8 represents contemporaneous equilibrium of the contemporaneous/cross-sectional coefficients added to the simplest and best fitting model from Models 1–7. Correlations among variables not shown for the sake of parsimony.
(TIF)

**S3 Fig. Trimmed summary of the standardized coefficient from the final model highlighting third to fourth year of college (Model 8, N = 751).** Only statistically significant paths are shown from coefficients in Table 2. All standardized structural coefficients ascertained from STDXY in Mplus as all variables were continuous. Underlined values represent stability coefficients; coefficients in standard text associated with predictors from the prior year in college are first-order cross-lagged coefficients, and coefficients associated with predictors from the current year of college are contemporaneous. The B-H FDR procedure was used to determine the statistical significance of all unstandardized coefficients. Based on the FDR procedure, all p-values less than .023 are reported statistically significant. $^{*}p\leq.023$, $^{**}p\leq.01$, $^{***}p\leq.001$.
(TIF)

**S1 Table. Summary of descriptive statistics and correlations between predictors, mediators, and outcomes across time.** Cronbach's alphas are presented on the diagonals. $^{*}p \leq .05$, $^{**}p \leq .01$, $^{***}p \leq .001$.
(PDF)

**S2 Table. Summary of longitudinal confirmatory factor analyses (N = 751).** T1 = pre-college, T2 = spring 1st year of college, T3 = spring 2nd year of college, T4 = fall 3rd year of college, T5 = spring 4th year of college. [a]The residual variance of item-2 at time-1 of the persistence intentions scale was not different from zero which causes convergence problems. Therefore, the variance of item-2 at time-1 was constrained to zero as were all residual correlations with the item. [b]The residual variance of item-2 at times 1, 2, & 3 of the persistence intentions scale was not different from zero which causes convergence problems. Therefore, the variance of item-2 at times 1–3 was constrained to zero as were all residual correlations with the item. $^{*}p \leq .05$, $^{**}p \leq .01$, $^{***}p \leq .001$.
(PDF)

**S3 Table. Summary of model fit and model comparison statistics for nested longitudinal mediation analysis models (N = 751).** Model 1 T-size results for equivalent testing were as follows (*a*) CFI = .970 and (*b*) RMSEA = .045; Model 1 T-size results relative to descriptive cutoff values were as follows for CFI (*a*) Excellent = .989, (*b*) Close = .950, (*c*) Fair = .920 and for RMSEA (*a*) Excellent = .022, (*b*) Close = .057, (*c*) Fair = .087; Model 8 T-Size results for equivalent testing were as follows (*a*) CFI = .946 and (*b*) RMSEA = .032; Model 8 T-Size results relative to descriptive cutoff values were as follows for CFI (*a*) Excellent = .985, (*b*) Close = .942, (*c*) Fair = .909 and for RMSEA (*a*) Excellent = .017, (*b*) Close = .055, (*c*) Fair = .088. $^{***}p \leq$ .001.
(PDF)

**S4 Table. Summary of standardized structural coefficients predicting engagement with social influence agents, social influence processes, and integration into STEM in the first-year of college (Model 8, N = 751).** T1 = pre-college, T2 = spring 1st year of college, T3 = spring 2nd year of college, T4 = fall 3rd year of college, T5 = spring 4th year of college. All standardized structural coefficients ascertained from STDXY in Mplus as all variables were continuous. Underlined values represent stability coefficients, coefficients in standard text associated with predictors from the pre-college are first-order cross-lagged coefficients, and coefficients associated with predictors from the 1st year of college are contemporaneous. The B-H FDR procedure was used to determine the statistical significance of all unstandardized coefficients. Based on the FDR procedure, all *p*-values less than .023 for unstandardized coefficients are reported statistically significant. $^*p \le .023$, $^{**}p \le .01$, $^{***}p \le .001$.
(PDF)

**S5 Table. Summary of standardized structural coefficients for social influence factors, social influence processes, and integration in the second-year of college (Model 8, N = 751).** All standardized structural coefficients ascertained from STDXY in Mplus as all variables were continuous. Underlined values represent stability coefficients, coefficients in standard text associated with predictors from the pre-college are first-order cross-lagged coefficients, and coefficients associated with predictors from the 1st year of college are contemporaneous. The B-H FDR procedure was used to determine the statistical significance of all unstandardized coefficients. Based on the FDR procedure, all *p*-values less than .023 for unstandardized coefficients are reported statistically significant. $^*p \le .023$, $^{**}p \le .01$, $^{***}p \le .001$.
(PDF)

**S6 Table. Summary of standardized structural coefficients for social influence factors, social influence processes, and integration in the third-year of college (Model 8, N = 751).** All standardized structural coefficients ascertained from STDXY in Mplus as all variables were continuous. Underlined values represent stability coefficients, coefficients in standard text associated with predictors from the pre-college are first-order cross-lagged coefficients, and coefficients associated with predictors from the 1st year of college are contemporaneous. The B-H FDR procedure was used to determine the statistical significance of all unstandardized coefficients. Based on the FDR procedure, all *p*-values less than .023 for unstandardized coefficients are reported statistically significant. $^*p \le .023$, $^{**}p \le .01$, $^{***}p \le .001$.
(PDF)

**S7 Table. Summary of standardized structural coefficients for social influence factors, social influence processes, and integration in the fourth-year of college (Model 8, N = 751).** All standardized structural coefficients ascertained from STDXY in Mplus as all variables were continuous. Underlined values represent stability coefficients, coefficients in standard text associated with predictors from the pre-college are first-order cross-lagged coefficients, and coefficients associated with predictors from the 1st year of college are contemporaneous. The B-H FDR procedure was used to determine the statistical significance of all unstandardized coefficients. Based on the FDR procedure, all *p*-values less than .023 for unstandardized coefficients are reported statistically significant. $^*p \le .023$, $^{**}p \le .01$, $^{***}p \le .001$.
(PDF)

**S1 File.**
(DOCX)

## Author Contributions

**Conceptualization:** Paul R. Hernandez.

**Data curation:** Michelle Williams.

**Formal analysis:** Paul R. Hernandez, David Loomis.

**Funding acquisition:** Michelle Williams, Crystal L. Park.

**Investigation:** Michelle Williams, Crystal L. Park.

**Methodology:** Paul R. Hernandez.

**Project administration:** Michelle Williams, Crystal L. Park.

**Supervision:** Michelle Williams, Crystal L. Park.

**Writing – original draft:** Paul R. Hernandez, Mica Estrada, David Loomis.

**Writing – review & editing:** V. Bede Agocha, Lauren M. Carney, Sharon Y. Lee, Michelle Williams, Crystal L. Park.

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
