## [Decision Letter · Decision Letter 0]

15 May 2020

PONE-D-20-08469

Testing Models of Reciprocal Relations between Social Influence and Integration in STEM across the College Years

PLOS ONE

Dear Dr. Hernandez,

Thank you for submitting your manuscript to PLOS ONE. After careful consideration, we feel that it has merit but does not fully meet PLOS ONE’s publication criteria as it currently stands. Therefore, we invite you to submit a revised version of the manuscript that addresses the points raised during the review process.

We would appreciate receiving your revised manuscript by Jun 29 2020 11:59PM. To enhance the reproducibility of your results, we recommend that if applicable you deposit your laboratory protocols in protocols.io, where a protocol can be assigned its own identifier (DOI) such that it can be cited independently in the future. For instructions see: http://journals.plos.org/plosone/s/submission-guidelines#loc-laboratory-protocols

We look forward to receiving your revised manuscript.

Kind regards,

Frantisek Sudzina

Academic Editor

PLOS ONE

Additional Editor Comments:

There are multiple ways to deal with the comments. Take time to think what would be the best option from your point view after reading the three reviews; I do not want to push you one way or the other.

Reviewers' comments:

Reviewer's Responses to Questions

**Comments to the Author**

1. Is the manuscript technically sound, and do the data support the conclusions?

Reviewer #1: Yes

Reviewer #2: Yes

Reviewer #3: Partly

2. Has the statistical analysis been performed appropriately and rigorously? 

Reviewer #1: Yes

Reviewer #2: Yes

Reviewer #3: Yes

3. Have the authors made all data underlying the findings in their manuscript fully available?

Reviewer #1: Yes

Reviewer #2: Yes

Reviewer #3: Yes

4. Is the manuscript presented in an intelligible fashion and written in standard English?

Reviewer #1: Yes

Reviewer #2: Yes

Reviewer #3: No

5. Review Comments to the Author

Reviewer #1: See my attached full review of the paper. My concerns are not with the way the study was conducted or analyzed. I do have concerns about the interpretation given to the model presented. In my attached review I have tried to explain the way in which i would interpret the findings. I think the research need to revise the final sections of the paper. They are in my opinion a bit too wedded to their original theoretical orientation and the data may not fully support their conclusions. I encourage the researchers to rethink those final sections.

Reviewer #2: This paper presents the results of a longitudinal study spanning the entirety of undergraduates’ careers and investigating the Tripartite Integration Model of Social Influences (e.g., how Social Influence Agents affect Social Influence Processes which may intern influence Social Integration and other relationships among these factors). Their results indicate that more diverse mentor networks and high quality faculty mentorship support and research engagement support STEM integration through (mediated by) their influence on science identity and science community values. In addition the authors find that high levels of initial integration predict pursuit of additional research experiences via the effect on science identity.

I find this paper to adhere to all standards and metrics for publication in PLOSone. Experiments and analyses are performed to a high technical standard, conclusions are appropriate and claims are aligned with results, and the study clearly identifies a gap in the literature. Furthermore, the writing is clear, concise, and while it is technically complex at times, I feel it would be accessible to a reader who is not overly familiar with SEM and the other techniques used.

Responses to the following minor comments would improve the paper:

Line 127. You have not yet discussed the process of reciprocal influence in the manuscript, clarify for what and whom the influence could be reciprocal.

Lines 214-218 It may be worth mentioning in this paragraph why you did not use the data set you drew on for this study, which includes HU students, to also analyze HU students and compare their outcomes to HO students. Was this a sample size issue?

Line 276 You explain that 22% of your HO sample were first-generation college students. In many studies, these students are reported as underserved or underrepresented. Could you comment on why these students are included in the HO sample?

Table 1. It may not be immediately clear to readers who do not frequently do statistics what N, M, and SD mean. Perhaps add these to the notes at the end of your table. In addition, please briefly explain how to interpret the numbers for N as these are not intuitive.

Lines 357-369. Please provide an explanation for why the pre-college survey responses and college survey responses differ substantially. It is clear why ‘university faculty’ are on one survey and ‘teachers’ are on the other, but it is not clear why family members and others were removed from the second scale.

Line 504 and Table 2. The p value cutoff you use in your table is 0.023 and it appears that research experiences did not make this cutoff for predicting scientific career persistence intentions as indicated by the absence of an * in the table (unless I am somehow reading the table incorrectly). Do you still want to discuss this result? ...or perhaps you just need to add the “*” in the table.

Table 2.

-Should the first listing of “Research Experiences” under T5 4th Year of College Outcomes actually be “Persistence Intentions”?

- It gets a little challenging to remember which components are “Social Influence Agents” and “Social Influence Processes.”. Could you somehow designate “agents” and “processes” in the table to remind readers of these components?

Lines 494-535

-It would be nice if you created a supplementary structural equation model figure that would present a schematic representation of the data in Table 2 and referenced it somewhere in your descriptive text. For readers who are helped by diagrams and schematics of data, this would elucidate relationships between variables. This figure would be a bit complex and large (hence the recommendation to include it in the supplement), but I feel it would help readers to interpret the results and view the data from a temporal perspective.

-It is a little counter intuitive having the contemporaneous coefficients discussed first when they are in the second part of the table. Could you cue the reader to look in the lower section of the table for these results in the text so that it is easier to cross-reference?

Line 615. Consider also referencing the earlier call: https://www.lifescied.org/doi/full/10.1187/cbe.11-03-0028

Lines 617-624 - Did you explore the possibility that influence operates more rapidly for HO students than for HU students? Did the prior literature that you refer to here explore HO or HU students? This may be worth commenting on.

Lines 626-644 - Do you have any hypotheses as to why values, specifically, are a mediator for HO students but not for HU students? Can you find or point to any literature that explores the differences between values and identity and discusses how these two social processes may be acting in order to explain your results?

Grammatical/Wording Suggestions:

Line 179 - It may make more sense to say ‘processes’ instead of ‘process’

Line 226 - Change ‘design’ to ‘designed’

Line 499 - Please add the word “and” between “self-efficacy” and “science identity”

Reviewer #3: This study examined longitudinal links among mentoring, psychological processes, and science career intentions among historically overrepresented university students. The paper has many strengths, and in particular I appreciated the rigorous mediation and model selection methods along with the ambitious goals to comprehensively test these processes. However, I have a moderate to serious concerns about the rationale for the study, the methods, and the results as outlined in detail below:

1. I wondered why the focus was on historically overrepresented groups. Prior studies having largely examined HU groups (lines 170-180) is not really an adequate rationale for the exclusion of HU students from the sample. Further, if an aim of the study is to identify subgroup differences/similarities (line 217-218), it makes more sense to include the subgroups in the current study. Currently, the study does not allow for direct comparisons, so differences between this study and prior studies could be attributed to the many differences in the data and study design that were earlier identified as unique strengths of this paper.

2. The rationale for the need and purpose of this study might be greatly strengthened by mentioning a specific theory or theories and their tenets rather than relying broadly on the propositions and potential extensions of “social psychology theories” (e.g., lines 69, 82, 666). Social cognitive models, for example, would be a more specific framework to draw upon, or expectancy-value theory in particular seems very relevant to this study.

3. I also have concerns about the measures. First, the reliabilities are a bit low for scientific career intentions. Second, why were measurement analyses presented for some but not all constructs (e.g., why was the faculty mentoring support measure not included in the CFAs or tests of measurement invariance)? Third, what was the rationale for the use of composite (mean) scores rather than latent constructs or factor scores? Lastly, all measures are self-report, introducing the strong possibility of over-inflated relations among variables due to common method bias.

4. Missing data presents another major concern. There is a great deal of missing data, on some variables greater than 80%. Importantly, there was no evidence presented for the MAR assumption (line 416) that must be met for FIML estimation to produce unbiased estimates. Reasons for missing data are clearly non-random as, for example, students did not complete the mentoring practices measure if they did not have a university faculty mentor. It does not seem appropriate to estimate relations of mentoring quality with the other variables for students who did not experience university faculty mentoring at all. Further, the available evidence (e.g., demographic percentages) does not support the claim that “participants largely mirror the demographics of the university” (line 259-260), particularly within this analytic sample (e.g., 58% female in the study vs. 51% female at the university; entire racial/ethnic groups were excluded).

5. The modelling approach is in need of more detailed explanation and rationale in the text. For example, it was unclear whether the many variables were modeled in separate models or together in one model, or whether this was the most appropriate approach given the potential for multicollinearity, higher-order factor structures, etc.

6. The results are very hard to understand because there are so many processes under consideration and because of the aforementioned lack of clarity in the modelling approach. I recommend more deliberately explaining each finding in both technical terms as well as plain-language interpretations of the meaning of the findings to help readers more easily digest the results. Due to the lack of clarity, it was difficult to determine how well the results support conclusions such as “social influence operates rapidly” (line 616).

7. What criteria (e.g., statistical test) were used to determine which is the “strongest” predictor (e.g., lines 502, 506, 544)?

8. The writing is generally strong, however a number of typos indicate the need for a thorough proofreading. For example: (line 191-192) “STEM majors with a larger and more diverse networks”; (line 208-209) “we hypothesized that the level faculty support”; (line 437) “persistence intensions”; (line 505-506).

6. PLOS authors have the option to publish the peer review history of their article (what does this mean?). If published, this will include your full peer review and any attached files.

Reviewer #1: No

Reviewer #2: No

Reviewer #3: No

---

## [Author Response · Author response to Decision Letter 0]

10 Aug 2020

Please see all responses provided in the attached Response to Reviewers document.

---

## [Editor Report · Decision Letter 1]

13 Aug 2020

Testing models of reciprocal relations between social influence and integration in STEM across the college years

PONE-D-20-08469R1

Dear Dr. Hernandez,

We’re pleased to inform you that your manuscript has been judged scientifically suitable for publication and will be formally accepted for publication once it meets all outstanding technical requirements.

Kind regards,

Frantisek Sudzina

Academic Editor

PLOS ONE

---

## [Editor Report · Acceptance letter]

20 Aug 2020

PONE-D-20-08469R1 

Testing models of reciprocal relations between social influence and integration in STEM across the college years 

Dear Dr. Hernandez:

I'm pleased to inform you that your manuscript has been deemed suitable for publication in PLOS ONE. Congratulations! Your manuscript is now with our production department. 

Kind regards, 

on behalf of

Dr. Frantisek Sudzina 

Academic Editor

PLOS ONE